Effects of delivery via pressure-adjustable pneumatic gas-powered dart gun of three antimicrobial drugs (ceftiofur crystalline free acid, tildopirosin, and tulathromycin) on drug disposition and meat quality in cattle

Hairgrove Thomas B. 1
http://orcid.org/0000-0003-0697-7149 Fajt Virginia 2 vfajt@cvm.tamu.edu
Gill Ronald 1
Miller Rhonda 1
http://orcid.org/0000-0002-1448-6289 Miller Michael 1 3
Mays Travis 4
1 Department of Animal Science, Texas A&M University , College Station, TX , United States
2 Department of Veterinary Physiology and Pharmacology, Texas A&M University , College Station, TX , United States
3 Current affiliation: W.H. Miner Institute , Chazy, NY , US
4 Texas Veterinary Medical Diagnostic Laboratory , College Station, TX , United States
Aly Sharif
Electronic publication date: 2021 Aug 4
Publication date: 2021
Volume: 9
Electronic Location ID: e11822
Received 2020 Feb 19; Accepted 2021 Jun 29
Copyright: © 2021 Hairgrove et al.
Copyright year: 2021
Copyright holder: Hairgrove et al.
License: This is an open access article distributed under the terms of the Creative Commons Attribution License, which permits unrestricted use, distribution, reproduction and adaptation in any medium and for any purpose provided that it is properly attributed. For attribution, the original author(s), title, publication source (PeerJ) and either DOI or URL of the article must be cited.
License URL: https://creativecommons.org/licenses/by/4.0/

Keywords: Pharmacokinetics, Pharmacology, Antimicrobial drugs, Remote delivery, Carcass quality

Funding: Texas Beef Council This work was supported by the Texas Beef Council. The funders had no role in study design, data collection and analysis, decision to publish, or preparation of the manuscript.

==============================
Background

Although Beef Quality Assurance guidelines do not recommend use of darting methods to deliver drugs, cattle in the US may be raised on farms and ranches without restraint facilities, and reports from the field suggest that dart guns are being used to deliver antimicrobial drugs. Few studies report whether this route of administration results in altered drug disposition or carcass quality.

Methods

Forty steers were blocked by sire and then randomly assigned to treatment with saline, ceftiofur crystalline free acid, tildipirosin, or tulathromycin delivered via dart gun. To assess drug disposition, eight ceftiofur, six tulathromycin, and six tildipirosin-treated calves were selected to measure plasma concentrations of drugs up to 10 days after drug administration. Steers were then fed a balanced ration for approximately 6.5 months and slaughtered. To evaluate carcass quality, tenderness of steaks from darted-side and non-darted sides was evaluated via Warner–Bratzler shear force testing. Due to the prohibition of extralabel routes of administration for ceftiofur in the U.S., animals treated with this drug did not enter the food supply.

Results

Ceftiofur disposition differed from published reports with lower mean Cmax but similar mean apparent elimination half-life. Tildipirosin disposition differed from published reports with lower Cmax and shorter apparent elimination half-life. Tulathromycin was similar to previous published reports but Cmax and apparent elimination half-life was highly variable. All steaks (from darted and non-darted sides) from cattle treated with ceftiofur and saline were more tender than from cattle treated with tulathromycin or tildipirosin (P = 0.003). There was a trend toward more tenderness in steaks from the non-darted compared to the darted side. Steaks from the darted side for one treatment, tildipirosin, were less tender than the non-darted side.

Conclusions

Pharmacokinetic parameters of ceftiofur crystalline free acid, tildipirosin, and tulathromycin to cattle using pressure-adjustable pneumatic gas-powered dart gun were estimated in this study. Delivery of tildipirosin and tulathromycin to cattle with dart gun may also result in detectable decreases in tenderness of harvested steaks.

Introduction

Stocker cattle are recently weaned calves that are placed on pasture and then allowed to grow to approximately 300 kg before being placed in a feedyard for the final phase of feeding prior to slaughter (Falconer & Anderson, 2006). Texas has historically been a large producer of stocker cattle, with Texas A & M AgriLife economists estimating 3.4 million stocker cattle in 2014 (S. Bevers, 2018, personal communication). Stocker cattle enterprises are more short-term than other phases of beef cattle production, with many producers not consistently engaged from year to year. Most antimicrobials used in stocker operations are prescription drugs and require veterinary oversight; however, anecdotal reports from stocker operators and agricultural extension personnel indicate that there is inadequate understanding of “extralabel use” and that drugs are commonly used in an unapproved and sometimes illegal manner. The National Cattlemen’s Beef Association has stated that: “The companies manufacturing, selling and promoting these methods of drug and product delivery have the responsibility and the obligation to develop data to establish efficacy, safety, animal welfare, food safety, and other concerns as compared to current BQA (Beef Quality Assurance) approved methods of drug/product administration. It is also possible that FDA approval may be required for drug delivery by these methods of injecting drugs/products and that issue needs to be addressed by the manufacturers. Until such time as this critical data becomes available these methods do not meet BQA injectable product administration guidelines.” However, anecdotal evidence suggests that darts are being used regularly, given that most stocker cattle are on pastures without working or capture facilities to use when treating cattle with injectable drugs. Given the few reports in the literature on the disposition of drugs administered via darts, the first objective of this study was to determine if the pharmacokinetics of three commonly used antibiotics are altered by using this route of administration. The antimicrobial products used in this study are single-dose injectable products approved for the treatment of pneumonia in cattle.

In addition to changes in drug disposition, previous studies have shown that injections lead to damage to the injected area, with the resultant remodeling and fibrous infiltrate leading to changes in muscle tenderness (George et al., 1996; George et al., 1995; Sullivan et al., 2009). Tenderness is an important contributor to consumer satisfaction of whole meat products such as steaks. Its importance is underlined by beef quality assurance programs that urge that all injectable products be administered subcutaneously rather than intramuscularly, when permitted on the drug label. The second objective of this study was to evaluate injection site reactions shortly after darting (13 days) and at the time of slaughter to determine if darting results in changes in meat quality and tenderness.

Materials and Methods

Animals and drug administration

A total of Forty steers were available from a project in which sires were genetically verified. A total of Forty Bos taurus-Bos indicus crossbred steers were purchased from the Department of Animal Science at Texas A & M University. Study design was a randomized block, with sires serving as blocks. The steers belonging to each sire were randomly assigned to one of four treatment groups, ten per group. There were ten sires among the 40 animals, with seven sires represented across all four treatment groups and three sires that were balanced across treatment groups as much as possible. Treatment groups were (1) 6.6 mg/kg ceftiofur crystalline free acid (CCFA, Excede, Zoetis Animal Health, Kalamazoo, MI, USA), (2) 4 mg/kg tildipirosin (TILD, Zuprevo, Merck Animal Health, Madison, NJ, USA), (3) 2.5 mg/kg tulathromycin (TULA, Draxxin, Zoetis Animal Health, Kalamazoo, MI, USA), and (4) 0.9% saline (SALN). Drugs were delivered by pressure adjustable pneumatic gas-powered dart gun (X-Caliper compressed air rifle, Pneu-dart, Inc. 15223 Route 87 Highway Williamsport, PA, 17701, USA). Based on the manufacturer’s recommendations, a pressure of 6.9 bars (100 psi) was used for all treatments except CCFA. Based on trial and error, CCFA was delivered using a pressure of 7.2 bars (104 psi) due to the weight of the dart slowing the velocity enough to prevent full penetration of the skin at the pressure used for the other drugs. A 16 gauge 1/2 inch single-port needle was used for all treatments, and animals were darted at a distance of 25 feet aiming for a region approximately 4″ below the tuber ischii in order to target the biceps femoris muscle (also known as the bottom round). Although Beef Quality Assurance guidelines recommend all injections be in the neck whenever possible, the location of injection for this study was selected based on anecdotal reports of how darts were being used in stocker cattle. In addition, it should be noted that the administration of CCFA in this manner would be considered illegal, since cephalosporins may not be administered via an extralabel route in the US (United States Food & Drug Administration Center for Veterinary Medicine, 2012), and CCFA is labeled for injection subcutaneously at the base of or the middle posterior of the ear (United States Food & Drug Administration Center for Veterinary Medicine, 2020). The actual route of administration cannot be determined when animals are darted, so some drug may be subcutaneous and some may be intramuscular. All darts appeared to completely expel their contents, but darts were not weighed before or after administration. Darts were collected after administration, and no needles appeared to remain in any animals.

A total of Five ceftiofur-, six tulathromycin-, and six tildipirosin-treated calves were randomly selected from each group of ten animals for sampling for plasma concentrations of drug. Three additional steers from the herd (in addition from the 40 described above) were administered ceftiofur as previously described at a separate time for pharmacokinetic assessments only to provide a better estimate of pharmacokinetic parameters and verify the marked variability in drug concentrations; no carcass quality or tenderness data were collected from these animals. Blood samples for drug analysis were collected at 0, 4, 8 h, and 1, 2, 3, 6 and 10 days after drug administration. Animal availability and financial considerations prevented the inclusion of animals injected via the conventional route of administration via syringe.

After treatment and blood collection, steers were assigned to one of three feedlot pens for feeding so that sire and treatment groups were balanced across pens. Steers were fed a balanced ration that included dry rolled corn, dried distillers’ grains, chopped alfalfa, molasses, and a mineral premix including lasalocid (1,320 g/ton, resulting in 33 g/ton on a dry matter basis or 30 g/ton at 90% dry matter as specified on the product label) for approximately 195 days (6.5 months). Steers were then slaughtered with captive bolt and exsanguination.

All animal procedures were approved by the Texas A&M University Institutional Animal Care and Use Committee (IACUC #2014-0316).

Drug analysis

All chemicals and reagents were obtained from VWR Scientific, Randor PA USA and ACS grade unless otherwise noted. D9-clenbuterol (C570000) and tulathromycin (T897150) were obtained from Toronto Research Chemicals, Toronto, Canada. Tildipirosin (sc-397306) was obtained from Santa Cruz Biotechnology. Ceftiofur (32442) was obtained from Sigma Aldrich. St. Louis, IL, USA. Reagent grade water (RG-water) was provided by in-house PureLab ULTRA system, ELGA Water Technologies LLC, Lowell, MA, USA.

For sample preparation, 1mL serum or plasma was aliquoted for analysis. Standard curves for the analytes of interest were prepared by fortifying the sample aliquots prior to Solid Phase Extraction (SPE) at concentrations of 0.5 to 1,000 ppb. D9-clenbuterol (40 µL) was added to all samples, calibrators and controls as an internal standard to improve the precision of the quantitative analysis. Tildipirosin and tulathromycin were analyzed by High Performance Liquid Chromatography Tandem Mass Spectrometry (LCMSMS) after isolation by SPE. SPE was processed using SPEWare CEREX48 (SPEWare Corp, Baldwin Park, CA, USA) processor and concentrator. A total of 3 mL/35 mg WWP SPE columns (SPEWare #676-0353) were used to extract the analytes of interest. SPE columns were preconditioned with 1 mL methanol, and 1 mL RG-water before application sample. Columns were washed with 1 mL RG-water prior to drying 10 min with nitrogen at 45 °C. SPE columns were eluted with 1 mL methanol before evaporation under nitrogen for 10 min at 45 °C. Residues were reconstituted with 100 µL 5% acetonitrile in RG-water prior to analysis by LCMSMS.

Ceftiofur and metabolites were analyzed after conversion to Desfuroylceftiofur Acetamide (DFCA). The residue from sample extract (from WWP) was added 450 µL 4% w/v Dithioerythritol in borate buffer pH 9 (4.75 g disodium tetraborate, and 925 mg potassium chloride in 250 mL RG-water). Samples were capped and heated at 50 °C for 15min. 200 µL Iodoacetamide solution 14% w/v in PO4 buffer (0.025 M, pH7) was added to the vials and then heated at 50 °C for an additional 15 min. Samples were cooled and 500 µL 1 N acetic acid was added prior to SPE. A total of 3 mL/35 mg Trace-B (SPEWare # 711-335M) were used to extract the DFC using the equipment mentioned above. Columns were preconditioned with 1 mL methanol, 1 mL RG-water, and 500 µL 1 N acetic acid. After sample application, columns were washed with 1 mL RG-water, 500 µL 1 N acetic acid and 1 mL methanol. Columns were then dried under nitrogen at 45 °C prior to elution with 2% ammonium hydroxide in methanol. Eluents were dried under nitrogen at 45 °C. Residues were reconstituted with 100 µL 5% acetonitrile in RG-water prior to analysis by LCMSMS.

High Performance Liquid Chromatography tandem Mass Spectrometry was performed on a TSQ Endura, Thermo Instruments, San Jose, CA, USA and Agilent 6410 Triple Quadrupole LC/MS. Analytes were separated on an Acentis Express C18 2.1 × 100 mm HPLC column (#53823-U, Supelco Inc. Bellefonte, PA, USA) using a mobile phase of 0.1% formic acid in both RG-water (solvent A) and LC-MS grade acetonitrile (solvent B). A solvent gradient was applied to the column at a flow of 300 µL/min of 5% B to 95% B over 8 min. The LCMSMS was run in electrospray mode with a 0.7 mass resolution on both quadrupoles.

Pharmacokinetic analysis

Noncompartmental analysis was performed to estimate the pharmacokinetic parameters in plasma for each individual animal for each drug (see Tables 1–3 for parameters).

Table 1 Estimated pharmacokinetic parameters for ceftiofur after single dose of 6.6 mg/kg ceftiofur crystalline free acid via dart (see Materials and Methods for explanation of darting).

Animal no.		599	608	638	645	680	1	2	3	Mean	SD	Median	
Cmax	ng/ml	155	16	241	238	174	134	17	41	127	93	145	
Tmax	h	8	8	4	8	8	24	24	24	*		8	
λz	/h	0.0057	0.0051	0.0226	0.0108	0.0091	0.0163	0.0211	0.0081	0	0	0	
t½λz	h	123	136	31	64	76	42	33	85	74	40	70	
AUC0-obs	ng * h/ml	4,096	898	6,854	4,161	2,645	4,783	704	2,158	3,287	2,085	3,371	
AUC0-inf	ng * h/ml	4,542	1,036	6,892	4,336	2,753	4,960	746	2,749	3,502	2,077	3,544	
AUC % Extrap		10	13	1	4	4	4	6	21	8	7	5	
AUMC0-obs	ng2 * h/ml	258,400	76,492	137,545	83,398	79,881	221,486	31,720	131,242	127,520	77,525	107,320	
AUMC0-inf	ng2 * h/ml	444,331	136,637	148,474	141,868	117,522	274,959	39,879	345,440	206,139	135,572	145,171	
MRT0-obs	h	63	85	20	20	30	46	45	61	46	23	46	
MRT0-inf	h	98	132	22	33	43	55	53	126	70	43	54	
Notes:

* Median is preferred for this parameter, so mean is not reported.

Tmax = time of peak serum drug concentration; Cmax = peak drug concentration; t1/2 = apparent elimination half-life, calculated as ln(2)/λz, λz being the first order rate constant associated with the terminal portion of the time-concentration curve as estimated by linear regression of time vs. log concentration; AUC0-obs = area under the time-concentration curve from time zero to last observed concentration calculated by the linear trapezoidal rule; AUC0-inf = area under the time-concentration curve from time zero extrapolated to infinity, calculated by adding the last observed concentration divided by λz to the AUC0-obs; AUMC0-obs = area under the moment curve from time zero to last observed concentration; AUMC0-inf = area under the moment curve from time zero extrapolated to infinity; MRT0-obs = mean resident time estimated using time zero to last observed concentrations, calculated as AUMC0-obs/AUC0-obs; MRT0-inf = mean residence time estimated using time zero to infinity, calculated as AUMC0-inf/AUC0-inf.

Table 2 Estimated pharmacokinetic parameters for tildopirosin after single dose of 4 mg/kg via dart (see Materials and Methods for
3 explanation of darting).

Animal No.		555	582	594	634	640	682	Mean	SD	Median	
Cmax	ng/ml	366	310	354	346	424	405	368	41	360	
Tmax	h	8	8	8	4	8	8	7	2	8	
λz	/h	0.0073	0.0075	0.0108	0.0099	0.0105	0.0026	0.0081	0.0031	0	
t½λz	h	95	92	64	70	66	267	109	79	81	
AUC0-obs	ng * h/ml	19,326	37,150	26,495	13,112	45,286	44,372	30,957	13,396	31,823	
AUC0-inf	ng * h/ml	25,433	43,122	28,009	14,725	49,426	10,0368	43,514	30,540	35,566	
AUC % Extrap		24	14	5	11	8	56	20	19	12	
AUMCo-obs	ng2 * h/ml	1,324,693	3,596,137	2,049,570	818,095	3,936,357	4,662,887	2,731,290	1,551,437	2,822,854	
AUMC0-inf	ng2 * h/ml	3,623,687	5,824,633	2,553,052	1,368,295	5,325,416	39,711,236	9,734,386	14,779,989	4,474,552	
MRT0-obs	h	69	97	77	62	87	105	83	16	82	
MRTo-inf	h	142	135	91	93	108	396	161	117	121	
Note:

See Table 1 for abbreviations.

Table 3 Estimated pharmacokinetic parameters for tulathromycin after single dose of 2.5 mg/kg via dart (see Materials and Methods for
3 explanation of darting).

Animal No.		569	637	646	670	671	672	Mean	SD	Median	
Cmax	ng/ml	544	797	618	1,221	421	109	619	374	581	
Tmax	h	4	4	48	24	8	8	16	17	8	
λz	/h	0.0109	0.0062	0.0081	0.0042	0.0070	0.0058	0.0070	0.0023	0	
t½λz	h	63	111	86	166	99	118	107	35	105	
AUCl\0-obs	ng * h/ml	41,330	711,43	89,334	186,274	52,928	7,906	74,819	61,196	62,036	
AUC0-inf	ng * h/ml	43,032	81,506	101,740	290,168	61,856	10,282	98,097	99,234	71,681	
AUC % Extrap		4	13	12	36	14	23	17	11	14	
AUMC0-obs	ng2 * h/ml	3,709,741	6,992,264	8,739,931	19,268,859	4,903,251	677,774	7,381,970	6,447,784	5,947,758	
AUMC0-inf	ng2 * h/ml	4,273,912	11,144,018	13,248,509	69,034,189	8,323,160	1,654,218	17,946,334	25,390,091	9,733,589	
MRT0-obs	h	90	98	98	103	93	86	95	6	95	
MRT0-inf	h	99	137	130	238	135	161	150	47	136	
Note:

See Table 1 for abbreviations.

Post-mortem carcass quality and tenderness assays

Steers were humanely slaughtered at the Rosenthal Meat Science and Technology Center, Texas A&M University, College Station, TX, USA. Carcasses were chilled for 48 h postmortem and the Bicep femoris from the right and left sides was removed for further evaluation. The Longissimus dorsi lumborum was also removed from one side of each carcass. Products from animals treated with CCFA did not enter the food supply.

Live Weight of the animal just prior to slaughter was measured. Dressing percentage was defined as hot carcass weight divided by live weight times 100, where hot carcass weight was determined prior to chilling. Carcasses were ribbed at the 12th and 13th rib interface. Marbling score, ribeye area (cm2), kidney, pelvic and heart fat (%), and subcutaneous fat thickness (mm) were determined (USDA, 2017). Yield grade and quality were calculated (USDA, 2017).

After USDA grade assessment, color CIE (International Commission of Light, Vienna, Austria) L * (lightness; 100 = white and 0 = black), a * (overall hue; 100 = red and 0 = green), and b * (overall hue; 100 =y ellow and 0 = blue) color space values were assessed (Minolta Colorimeter, CR-300, 8 mm diameter head, 10° standard observer, D65 light source; Minolta Co., Ramsey, NJ, USA). White and black tiles were used daily to calibrate the Minolta Colorimeter. At the 12th rib Longissimus dorsi interface, three values were obtained per carcass and averaged. A total of three random pH values were obtained from the same location using a puncture pH probe (Hach, H100, Loveland, CO, USA) and averaged. This value was defined as the final carcass pH. The pH meter was calibrated (pH 4.0 and pH 7.0 standard buffers) at the start of each day. Color and pH were determined on the same lean surface as used for grade assessment. Hump height measurement of the Rhomboideus muscle (hump) was taken at its highest point.

The presumed site of dart entry was identified via visual examination for increased connective tissue or marbling or by digital palpation for slightly thicker, denser areas of muscle by a trained veterinarian with no knowledge of treatment. From the darted side in the bottom round muscle, 5–2.54-cm bottom round steaks were cut, the first steak included the presumed site of the dart entry (Position 1), and then two proximal (Positions 2 and 3) and two distal (Positions 4 and 5) 2.54 cm steaks were removed from the presumed site of dart entry. One corresponding 2.54 cm steak from the control side (Position 6) was also collected in the similar anatomical location from the dart side.

From the Longissimus dorsi lumborum muscle, beginning at the anterior edge, 4–2.54 cm top loin steaks were cut, vacuum-packaged and randomly assigned to aging for 1, 7, 14, or 21 days at 4 °C (labeled as Positions 1–4 in Table 4). Bottom round steaks were vacuum-packaged and aged for 14 days. For Warner–Bratzler shear force determination, top loin and bottom round steaks were weighed, iron constantan thermocouples (TT-J-36-SLE, Omega Engineering, Inc., Stanford, CT, USA) were inserted in the geometric center of each steak, and steaks were cooked on flat top electric skillets (Hamilton Beach grill, Hamilton Beach/Proctor-Silex, Inc., Southern Pines, NC, USA) to a final internal temperature of 70 °C. Internal cook temperature was monitored with hand-held recorders (model HH-21, Omega Engineering, Inc., Stanford, CT, USA). Steaks were turned after reaching 35 °C internal temperature. After cooking, cooked weight was recorded and cook yield was calculated (Cook Yield = 100 − ((Cooked Chop Weight/Raw Chop Weight) * 100).

Table 4 Least squares means for Warner–Bratzler Shear Force (kg) for top loin and bottom round steaks.

Treatment	Top loin steaks	Bottom round steaks	Bottom round steaks	
Dartedle	Controle	
Dart treatment	0.21d	0.003d	0.08d		
CCFA	3.04	3.10a	3.22ab	3.10a	
SALN	2.77	3.11a	3.27ab	3.11ab	
TILD	3.19	3.44b	3.71b	3.08a	
TULA	3.00	3.45b	3.51ab	3.32ab	
Position	0.73d	0.48d			
1	3.05	3.43			
2	3.06	3.27			
3	3.02	3.16			
4	2.86	3.36			
5		3.28			
6		3.15			
Age day	<0.001d				
1	3.85c				
7	3.11b				
14	2.67ab				
21	2.36a				
Root mean square error	0.838	0.693	0.672		
Notes:

abcMean values within a column and effect followed by the same letter are not significantly different (P > 0.05).

dP value for the treatment effect from the Analysis of Variance table.

eControl = right side, not darted; Darted = left side, darted from the steak removed where the dart insertion point was identified.

Positions in top loins steaks (see text for details): 1 = steak from anterior edge of Longissimus dorsi lumborum muscle; 2 = steak immediately posterior to 1; 3 = steak immediately posterior to 2; 4 = steak immediately posterior to 3.

Positions in bottom round steaks (see text for details): 1 = steak from presumed site of dart entry, 2 = 1st steak distal to presumed site of dart entry, 3 = 2nd steak distal to presumed site of dart entry, 4 = 1st steak proximal to presume site of dart entry, 5 = 2nd steak distal to presumed site of dart entry, 6=steak from non-darted (right) side at similar location to the darted side.

Cooked steaks were chilled for 24 h and then one 1.27 cm diameter core, parallel to the longitudinal orientation of the muscle fibers, was removed from six-seven positions designated on each steak. Steaks were assigned codes prior to cooking and laboratory personnel were blinded to treaments. Cores were sheared once with Warner-Bratzler shearing device (United Smart-1 Test System SSTM-500, United Calibration Corp., Huntington Beach, CA, USA; 200 kg load cell, 200 mm/min head speed) certified by United Testing Systems, Inc. using a Warner–Bratzler stainless steel blade (1.168 cm thick). Mis-shaped cores were not used. The maximum force in kg for cores from one steak were averaged.

Data analysis

Carcass grade data, color, pH and Warner-Bratzler shear force measures were analyzed using Analysis of Variance with the Proc GLM procedure of SAS (v9.4, SAS Institute, Cary, NC, USA) with an alpha of < 0.05. For grade data, color and pH, carcass was defined as the experimental unit. The dart treatment was defined as a main effect and the pen cattle were fed in was defined as a fixed effect. Warner–Bratzler shear force data were analyzed for top loin steaks. Main effects were defined as dart treatment, steak position, aging day and their subsequent two-way interactions and pen fed was used as a fixed effect. For bottom round steaks, data were analyzed with dart treatment and position of steak from the injection site, and their interaction as a main effects. The pen cattle were fed in was also defined as a fixed effect. A third analysis for bottom round steaks compared sides within a carcass for Warner–Bratzler shear force. The undarted side was defined as a control and the darted side was defined as the treatment. For all analyses, least squares means were calculated. Differences in least squares means were determined using the Tukey–Kramer function within SAS for multiple mean comparisons.

Results

Drug concentrations and pharmacokinetics

Mean drug concentrations at sampling time points for the 3 drugs are shown in Figs. 1–3, and mean and median pharmacokinetic estimates are in Tables 1–3.

Figure 1 Mean serum concentrations of ceftiofur after single dose of 6.6 mg/kg via dart (see text for explanation of dosing) in eight animals.

Error bars are standard deviation.

Figure 2 Mean serum concentrations of tildopirosin after single dose of 4 mg/kg via dart (see text for explanation of dosing) in six animals.

Error bars are standard deviation.

Figure 3 Mean serum concentrations of tulathromycin after single dose of 2.5 mg/kg via dart (see text for explanation of dosing) in six animals.

Error bars are standard deviation.

Carcass quality and tenderness

Carcass quality and yield factors by dart treatment indicated that dressing percentage and L* values were significantly different (P = 0.05 and P < 0.001, respectively) across treatment groups (Table 5). Dressing percentage was lower (P = 0.05) in cattle treated with SALN than cattle treated with CCFA and TILD. CIE color space values indicated L* values were significantly higher (P < 0.001) in cattle treated with CCFA, TILD and TULA resulting in lighter lean color than the cattle treated with the SALN control. Dart treatments had no effect on other carcass quality and yield grade factors.

Table 5 Least squares means for carcass quality and yield factors by dart treatment.

Treatment	Live weight (lbs)	Dressing %	Marbling score	Fat thickness (in)	Ribeye area weight (in2)	Hot carcass (lbs)	KPH %c	Yield grade	CIE color space values	pH	Hump (in2)	
L*	a*	b*	
P value	0.98	0.05	0.60	0.88	0.59	0.92	0.83	0.88	<0.001	0.10	0.16	0.22	0.86	
CCFA	1,097.7	66.2b	409	0.51	12.97	728.0	2.21	2.88	44.23b	14.32	7.03	5.57	4.36	
SALN	1,092.0	64.5a	408	0.55	12.91	704.0	2.36	2.95	40.07a	15.37	7.43	5.56	4.51	
TILD	1,084.1	66.7b	402	0.57	13.46	722.8	2.33	2.86	44.48b	13.19	6.24	5.60	4.44	
TULA	1,079.7	65.8ab	434	0.60	12.75	710.9	2.36	3.13	45.13b	14.49	7.09	5.53	4.23	
RMSE	114.57	1.66	56.60	0.24	1.20	82.41	0.42	0.85	1.98	1.80	1.14	0.08	0.73	
Notes:

abMean values within a column and effect followed by the same letter are not significantly different (P > 0.05).

cKPH % = Kidney, Pelvic and Heart Fat

As expected, top loin steak Warner-Bratzler shear force values across all treatments and positions were lower (P < 0.001) at 14 and 21 d age postmortem than at 1 and 7 d aging times (Table 4). Shear force values were also lower (P < 0.001) in steaks aged 7 d when compared to steaks 1 d postmortem. Neither the dart treatments (P = 0.21) nor position within the strip loin (P = 0.73) affected Warner–Bratzler shear force values.

There was no difference (P > 0.05) among core sample tenderness within steaks. There was also no difference in tenderness between steaks collected at the various positions within the bottom rounds (Table 4). All steaks (from darted and non-darted sides) from cattle treated with ceftiofur and saline were more tender than from cattle treated with tulathromycin or tildipirosin (P = 0.003).

There was a trend (P = 0.08) toward more tenderness in steaks from the non-darted compared to the darted side (Table 4). Steaks from the darted side for one treatment, tildipirosin, were less tender than the non-darted side (P < 0.05).

Discussion

Drug concentrations and pharmacokinetics

It is important to acknowledge that a limitation of this study is the lack of direct comparisons to conventionally injected antimicrobial drugs. Cattle availability and financial constraints prevented us from including additional animals in our study, and future studies could provide controlled comparisons of drug disposition after different routes or methods of administration. The following discussion reviews comparisons to previously published estimates of the pharmacokinetics of the three drugs after conventional subcutaneous injections. These comparisons are acknowledged to be subjective in nature, and pharmacokinetic parameters can be variable across studies and across individual animals.

Ceftiofur crystalline free acid

In this discussion, ceftiofur includes the combination of parent drug and its derivates, mainly desfuroylceftiofur or desfuroylceftiofuracetamide; desfuroylceftiofur is microbiologically as active as parent drug against cattle pneumonia pathogens, and desfuroylceftiofuracetamide is derived during sample processing from desfuroylceftiofur. Concentrations of ceftiofur reached a much lower Cmax than previously reported after conventional injection in Holstein calves (4,260 ng/ml in plasma) (Foster, Martin & Papich, 2016), in dairy cows (4,150 ng/ml in serum) (Witte et al., 2011), in beef calves (6,020 ng/ml in plasma) (Washburn et al., 2005), and in neonatal Holstein calves (3,230 ng/ml) (Woodrow et al., 2016). This is in sharp contrast to the mean Cmax in the present study of 127 ng/ml (range 16–238 ng/ml).

On the other hand, the median observed Tmax of 8 h was similar to previous reports: 12 h (median) (Woodrow et al., 2016), 13.1 h (mean) (Foster, Martin & Papich, 2016), 24 h (Witte et al., 2011), although in the last study, samples were only collected at 2, 24, and 18 h (back-transformed least squares means) (Washburn et al., 2005). This suggests that ceftiofur is being absorbed at a similar rate after darting but not to a similar degree as injecting in the ear, the route of administration indicated on the label. It is also important to reiterate that darting CCFA products is illegal in the US, because cephalosporin use at an extralabel regimen in cattle is prohibited (United States Food & Drug Administration Center for Veterinary Medicine, 2012).

Mean elimination half-life (74 h, range = 31–136 h) was in the range of previously reported times of 61 (Woodrow et al., 2016), 62 (Washburn et al., 2005), and 104 h (Foster, Martin & Papich, 2016). This suggests that the most important difference between darting and conventional injection is the amount of drug that enters the animal, although the range of estimated half-lives in this study might result in significant variability in estimates of meat withdrawal times via this route of administration. Tissues were not tested for ceftiofur concentrations in this study, however, so this is only speculation.

It should also be noted that, regardless of potential issues with therapeutic effectiveness and impact on meat withdrawal time, administering CCFA at any other site than the base of or middle posterior portion of the ear in a subcutaneous manner would be considered illegal in the US because of restrictions on extralabel use of cephalosporins (United States Food & Drug Administration Center for Veterinary Medicine, 2012).

Tildopirosin

Only one previous report of the pharmacokinetics of tildopirosin in cattle has been published (Menge et al., 2012). Mean Cmax was 711 ng/ml, and mean elimination half-life was 210 h. These values are quite different from the present study: mean Cmax was 368 ng/ml, mean elimination half-life was 109 h (see Table 3). Several factors could help explain the differences: blood sampling was performed more intensely early in the dosing period in the previous study (0.5, 1, 2 h) with the Cmax being reported at a mean Tmax of 0.7 h compared to the present study (first sample at 4 h). It is quite possible that a higher Cmax would have been observed if we had collected earlier samples. In the previous study, samples were collected up to 21 days after drug administration, whereas we only collected for 10 days. Without access to the raw data from the previous study, we cannot know for sure, but visual examination of the graph of plasma drug concentrations over time suggests that the slope of the elimination curve is not considerably different from 2–10 days as compared to 10–21 days. Since this slope is the main contributor to the estimate of elimination half-life, we speculate that our estimates would not have changed significantly if we had collected samples for longer, but we cannot know for sure. A higher than desirable percent of AUC was extrapolated to infinity in the present study, that is, 24% in one animal and 56% in another. In all other animals (n = 4), average percent of AUC extrapolated to infinity was 10%, suggesting that blood samples were collected for a sufficient amount of time in those animals to adequately characterize pharmacokinetic values. These data do suggest, however, that darting of tildopirosin could affect the disposition of the drug in cattle.

Tulathromycin

Mean Cmax (619 ng/ml) in the present study was in the range of previous reports of conventionally injected tulathromycin: 277 ng/ml (Cox et al., 2010), 489 ng/ml (Nowakowski et al., 2004), and 414 ng/ml (Nowakowski et al., 2004), and 1,820 ng/ml (Foster, Martin & Papich, 2016). Perhaps the most relevant comparison is to Holstein calves administered tulathromycin via pneumatic dart, in which the mean Cmax was 206 ng/ml (Coetzee et al., 2018). It should be noted, however, that in that study, there was not a significant difference between Cmax after conventional injection and successful dart delivery of tulathromycin (Coetzee et al., 2018), largely due to the high variability in individual Cmax, but there was a difference in Cmax between successful and unsuccessful darting. In the present study, we did not appreciate any failure in dart delivery of drug, but mean Cmax observed in the present study suggests that darting may result in enough variability in drug concentrations to alter estimates of meat withdrawal times.

Estimated apparent elimination half-life in the present study (107 h) is similar to the previous report of 111 h in darted cattle (Coetzee et al., 2018). This estimate is somewhat longer than reported elimination half-life after conventionally injected tulathromycin in other studies in cattle (92 (Nowakowski et al., 2004), 81 (Foster, Martin & Papich, 2016), and 64 h (Cox et al., 2010)). However, sampling times were not as prolonged in our study as in two of the previous studies. One collected samples for 336 h (Nowakowski et al., 2004) and the other 360 h (Cox et al., 2010), which could have impacted the estimation of elimination half-life.

Carcass quality and tenderness

We are not aware of previous reports on the effects of remote delivery devices on tenderness at harvest, although a recent study of pneumatic darts similar to those used in the present study evaluated muscle damage by quantification of serum creatine kinase up to 6 days after injection of tulathromycin via dart (Coetzee et al., 2018). In that study, no difference was detected in creatine kinase concentrations between dart-injection and hand-injection of tulathromycin, but sham-injected animals were not included as a comparison in the study. In other studies, gross lesions have been reported at injection sites after intramuscular injection, including two reports of remote delivery devices that differ from the one used in the present study. Investigators in Canada used a crossbow to deliver oxytetracycline and tilmicosin to the neck or round (exact location not specified) region in cattle 30 days from slaughter, and at slaughter they noted lesions in darted and hand-injected cattle, including granulation, neovascularization, and abscesses (Van Donkersgoed et al., 1999). In the other darting study, the drug was delivered in patented devices called biobullets, which were designed to dissolve once delivered into the muscle (Edwards & Stokka, 1993). In that study, some of the biobullets did not enter the skin properly, and the ones that did enter the skin caused significant tissue damage when the animals were necropsied 2 h after injection. In the present study, we did not evaluate tissue damage in the early days after injection except during gross examination of the pilot study animals, so it is unknown if muscle was damaged at dart delivery during the study. However, the only statistically significant differences at slaughter in tenderness as evaluated via the Warner–Bratzler method (see Table 4) were in bottom round steaks from tildipirosin-and tulathromycin-darted animals as compared to saline-and ceftiofur-darted animals. Consumers can detect differences of 0.5 kg in Warner–Bratzler tenderness measures (Miller et al., 2001), so the statistically significant differences noted between tildipirosin-darted and non-darted steaks (Table 4) are potentially detectable by the consumer. The differences between ceftiofur-and saline-treated as compared to tildipirosin-or tulathromycin-darted animals (see Table 4) may not be detectable by the consumer even though statistically significantly different.

Conclusions

Delivery of ceftiofur crystalline free acid, tildipirosin, and tulathromycin to cattle using pressure-adjustable pneumatic gas-powered dart gun may result in pharmacokinetics and therefore meat withdrawal times and therapeutic efficacy that are different from those expected after conventional injection of these drugs. However, the most notable differences from published pharmacokinetic variable estimates as well as variability was with the ceftiofur crystalline free acid-treated animals. Delivery of tildipirosin and tulathromycin to cattle with dart gun may also result in detectable decreases in tenderness of harvested steaks. Delivery of ceftiofur crystalline free acid in this manner would be considered an illegal extralabel use and is therefore discouraged.

Supplemental Information

Supplemental Information 1 Weights of cattle as evaluated weekly during the study.

Weights are in pounds.

Click here for additional data file.

Supplemental Information 2 Weight at slaughter of cattle and their and carcass and meat quality characteristics.

“Carcass data” includes yield and quality grading data of steaks cut as described in the manuscript; “Cook data” includes steak characteristics; “WarnerBratzler Data” includes tenderness assessments for all steaks cut as described in the manuscript.

Click here for additional data file.

Supplemental Information 3 Serum concentrations in individual cattle of three antimicrobials after darting.

“Cef data” are serum concentrations after administration of ceftiofur crystalline free acid; “tildo data” are serum concentrations after administration of tildopirosin; and “tulath data” are serum concentrations after administration of tulathromycin as described in the article.

Click here for additional data file.

The authors acknowledge Crystal Waters for the collection and analysis of tissue samples and initial tenderness data analysis, Cory Langston for discussions about the pharmacokinetic analysis, and Dan Hale and Davie Griffith for their assistance with study design and sample preparation and handling.

Additional Information and Declarations

Competing Interests

Author Contributions

Animal Ethics

Data Availability

The authors declare they have no competing interests.

Thomas B. Hairgrove conceived and designed the experiments, performed the experiments, authored or reviewed drafts of the paper, and approved the final draft.

Virginia Fajt conceived and designed the experiments, analyzed the data, prepared figures and/or tables, authored or reviewed drafts of the paper, and approved the final draft.

Ronald Gill conceived and designed the experiments, performed the experiments, analyzed the data, authored or reviewed drafts of the paper, and approved the final draft.

Rhonda Miller performed the experiments, analyzed the data, prepared figures and/or tables, authored or reviewed drafts of the paper, and approved the final draft.

Michael Miller performed the experiments, authored or reviewed drafts of the paper, and approved the final draft.

Travis Mays performed the experiments, authored or reviewed drafts of the paper, and approved the final draft.

The following information was supplied relating to ethical approvals (i.e., approving body and any reference numbers):

Texas A & M University Institutional Animal Care and Use Committee approved the animal use protocol for this study (IACUC #20140316).

The following information was supplied regarding data availability:

The raw data are available in the Supplemental Files.

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
