# Peer review of "Effects of delivery via pressure-adjustable pneumatic gas-powered dart gun of three antimicrobial drugs (ceftiofur crystalline free acid, tildopirosin, and tulathromycin) on drug disposition and meat quality in cattle"

_PeerJ, doi:10.7717/peerj.11822_

## Round 0.1 · original submission · Major Revisions

Thank you for submitting your manuscript, in addition to addressing the reviewers' thorough comments on your manuscript please clarify the following:

1- whether the ceftiofur crystalline free acid (CCFA) used conforms to the formulation, composition and concentration of CCFA available on the market and labeled for subcutaneous injection in the posterior aspect of the ear’s attachment to the head.
2- The route of administration of CCFA should also be explicitly mentioned, it is assumed to be intramuscular (since the biceps femoris was the target).
3- The fate of the steers treated with CCFA should be mentioned, were they disposed off or did they enter the food chain?

This information is needed to clarify if the CCFA trial arm treatments were under extralabel use, if they were then justification is needed for not following the federal law(1) prohibiting extralabel use of cephalosporins (with exception of cephapirin).

For example, was the trial designed and implemented for non-US cattle – since the federal law applies to the US only and European countries for example do not have this restriction? What was the fate of these animals - were they disposed off, or what precautions were made to avoid them entering the food chain? Was an elongated withdrawal period observed based on a consult with FARAD or FDA itself?

The discussion section should address the above concerns and describe the justification for the CCFA arm of the trial, discuss the respective FDA law so it is clear that extralabel use of Cephalosporins including CCFA (and with exception of Cephapirin) is illegal in the US when a different dose or route are used in major food animal like cattle.

Minor comment: The dose for all 3 treatments should also be described as it is missing from lines 118 to 119, also it is not clear what is meant in line 120 “3 additional steers that were administered ceftiofur at a separate time” please reword for clarity.



1 Cephalosporin Order of Prohibition Questions and Answers, Jan 6th 2012, US FDA. https://www.fda.gov/animal-veterinary/antimicrobial-resistance/cephalosporin-order-prohibition-questions-and-answers

Reviewer 1 ·

Basic reporting

- The article is written in clear language, and the structure of the submitted article conforms to the suggested format. The introduction clearly states the objectives of the study and places the study in context. The following specific lines/tables are noted for minor corrections, which would improve the paper’s clarity.

75-76: There is a statement here that should be referenced.

123: If known, the concentration of Lasalocid in the final TMR should be given, rather than the concentration in the pre-mix.

131: The use of D9-Clenbuterol in this study should be explained (is this a laboratory control?)

198-200: The number of cuts made is not easy to understand based on these sentences. Were there five cuts made on the darted side and one on the non-darted side?

249, 252 and Table 5: These both mention the testing of striploins. Were striploins tested or bottom rounds? There is no mention of collecting striploins in the experimental design.

273-274: A mean value is given (0.13mcg/mL) that is not within the range of individual values given (0.160-0.238mcg/mL). Using the table it appears that the min should be 0.016 and max 0.241.

Table 4: For the values L*, a*, and b*, it appears that the same asterisk used for KPH applies to these values, a different symbol should be used for KPH.

Table 7: Is the “dart” category just one value (that of the darted steak) or an average of all 5 steaks on the darted side?

Figures 1-3 and Tables 1-3: These all state in the title “See text for explanation of dosing”, yet an explanation of dosing was not easily found in the text. A specific page, line, or section reference should be made.

No specific location:
- The format for reporting p values should be consistent throughout
o Italicized vs normal text
o Spaces between the p, the sign, and the number vs no spaces
o Same number of significant digits throughout
- Would it be possible to combine tables 6 and 7, maybe even table 5?
- In the tables the treatments are inconsistently labeled (some spelled out, others abbreviated)

Experimental design

- The research question is clearly defined, and is relevant and meaningful. The knowledge gap being investigated is identified.

- The methods are consistent with ethical standards in bovine veterinary research; however, the illegal use of CCFA should be more clearly addressed as to why it was included in the study, reiterate the fact it’s illegal, describe whether these animals entered the food supply, etc.

- There are some areas of the methods section where more detail is required, both to provide confidence in the technical standard of the research, to promote reproducibility, and allow readers to appropriately interpret the results.

o Overall, there appears to be sufficient detail for reproducibility in the laboratory methods (both pharmacokinetic and post-mortem carcass quality and tenderness assays), however, more detail in the methods section regarding the live animal phase of the experiment would be useful. It would be difficult to reproduce the live animal phase from the current text.

o The description of the blocking of animals and assignment to treatment groups is well done, however it is unclear to the reader how the final numbers in the pharmacokinetic study were reached. It is evident that more animals were randomly assigned to treatment groups than were analysed - forty steers were randomly assigned, yet only 5-6 per drug treatment, for a total of 17, had blood samples measured. It is unclear how this subset of animals was selected for analysis; were they randomized? Additionally, 3 animals were injected with ceftiofur at a later time, but the reason is not explained in the manuscript. More details should be provided in the methods and results section outlining the criteria used to select animals for analysis, and details should be provided about the origin of the additional 3 animals in the ceftiofur group, as well as the rationale for adding them to the study.

o For drug delivery, some details are provided on the range, needle type, and intended target, but more detail would be required to reproduce the experiment. A good example of a methods section that provides this level of detail can be found in Coatzee et. al. (referenced in the study). For examples of some questions while reading the manuscript: What was the gun’s model and what pressure was it charged to? Were darts recovered and checked for discharge of their contents? Did any darts hit the animal in an unintended location, and if so, was this animal still used for either pharmacokinetic or meat tenderness analysis? Did any needles remain lodged in the tissue of an animal, and if so, for how long, and what was the procedure for removing them? If available, providing these details would aid reproducibility and improve clarity. If not available, the authors should state that they are unknown or not recorded.

o Drugs and drug doses, while listed in the respective tables, should also be listed in the methods section. Rather than saying “Each drug was administered to 10 steers…” on line 114, the authors need to actually describe the experimental groups as that is fundamental to understanding the study.

o With regards to identifying the injection site for analysis of the meat (line 197), no mention is made of the observer being blinded to the treatment. If the observer was blind to the treatment group, this should be stated to improve confidence that no bias was inadvertently introduced in selection of the meat samples. If the laboratory personnel processing the meat and performing the shear force testing were blind to the treatment group, this should also be stated.

Validity of the findings

- The conclusion of the study is: “Delivery of ceftiofur crystalline free acid, tildipirosin, and tulathromycin to cattle using pressure-adjustable pneumatic gas-powered dart gun may result in altered pharmacokinetics and therefore altered meat withdrawal times and altered therapeutic efficacy. Delivery of tildipirosin and tulathromycin to cattle with dart gun may also result in detectable decreases in tenderness of harvested steaks” (lines 363-367). This conclusion is reasonably supported by the data, however, it is recommended that the following points are addressed in both the discussion and conclusion sections:

o The experiments are not controlled in a way that allows the pneumatic dart delivery method to be compared with conventional injection methods for the same drugs, since there are no conventionally injected animals in this study to serve as positive controls. The authors do acknowledge that there are several reasons, including dose and sampling time, that could account for the differences between their research and existing published studies (lines 297-299), but more information should be provided in the discussion section about the reason(s) for not including these controls in the study, the limitations of comparing data obtained in this study with that of previous studies, and opportunities for future research.

o Recommend more information in the discussion section comparing the methods used in this study with those of the previous pharmacokinetic studies that are referenced in this manuscript.

o Since the study design does not allow for direct comparison between conventional and dart delivery injection methods, the conclusions should avoid comparative terms such as “altered”, or should state that meat tenderness/pharmacokinetic data may be altered when compared with existing data from previous studies.

- Since the injection site recommended by the BQA is in the neck, more information in the discussion section about the rationale for performing injections in the bottom round would be very informative. For example, was this site selected because it is this a common site of injection with pneumatic darts among producers, in the authors’ experience?

- A brief discussion of how the results from the pilot study informed the design of the main study would also aid the reader’s understanding of this paper, particularly with respect to the use of flunixin. Without discussion, it is not clear to the reader why this drug was used. A short section on the pilot study should be included in the discussion.

- Related to the above point, the second objective of the study (evaluation of injection site reactions at 13 days) seems to have been completed in the pilot study only. Therefore, a discussion of how these results informed the design of the main study would be very informative.

·

Basic reporting

The article is overall well written and worthy of publication (after corrections and improvements). Adequate background information is given to demonstrate that the information conveyed in this manuscript is valuable to the beef industry and veterinary profession. There are several areas where clarity could be improved, some portions where additional information is needed, and at least one notable typographical error.

Specific suggestions or points of concern:
Line 72: Would suggest clarifying “The ANTIMICROBIAL products used in this study….”
Line 75: “….previous studies have shown that…..” Provide citation for these studies.
Line 79: “… urge that all injectable products should be administered subcutaneously rather than intramuscularly” (I suggest adding “if permitted by drug label”).
Line 115: Device is described as “pressure adjustable.” Please describe how the pressure to be used was determined, and whether it was consistent across the study.
Line 138: Reference made to SPE prior to the acronym being defined (line 141).
Line 229: It is unclear why results for drug concentrations are not reported in results, but rather primarily included in the discussion.
Line 273-274: Statement is made that mean Cmax is 0.13, with a range of 0.160 to 0.238. The mean isn’t within the reported range. Looking at table 1, it appears the range should be 0.016 to 0.238.
Line 277: Statement of …”24 hrs but samples only collected at 2 and 24 hr….” is confusing.
Line 297: It is stated the other study sampled more intensely early in the dosing period. Did Cmax occur during those early sampling intervals?
Line 330: “…but sampling times were different (in the present study?)….”
Various: Interpretation of “similarity/dissimilarity to other publications” and “widely variable” seems subjective and perhaps inconsistent. For example, line 274: “…wide range of individual values.” This reviewer doesn’t view that range as particularly wide. The assessment that median Tmax was similar to other reports is open to interpretation- 8 hours seems notably different than 18 hours (Washburn) or 24 hours (Witte- although it is noted that sampling was only done at 2 and 24 hours). This is apparent again in lines 314-317, where tulathromycin is in the range of previous reports, where previous reports range from 277 to 1820. Such a range seems quite broad, and thus it is expected that the current study would fall within this range. If 0.160 – 0.238 is considered a wide range of individual values (line 274), then 277 to 1820 should likely be identified as extremely broad, and effort made to explain why such a range should exist, and the implications of such for the current study [as noted above, the range of 0.16 to 0.238 is apparently in error].

Table 1: Various terms in the legend don’t correspond to the terms in the table. For example, table includes the term “AUClast”, which would seem to correspond to the description in the legend for “AUC0-obs.” Similarly, there is no description of the term “AUCINF_obs”, which is used in the table. This occurs for nearly all of the terms.
Discussion information for ceftiofur is reported as mcg/ml in the text, but as ng/ml in the table.

Experimental design

It is unclear what inclusion of the pilot study contributes to the manuscript. Of particular concern is the appropriateness and relevance of administering 10 ml flunixin meglumine. This is a very large volume (exceeding even the high end of the label dose) of a well-known irritating compound. Also questionable is the use of 5 ml of CCFA at a dose at or below label for these animals. Additional information and context should be provided if the pilot study is to remain part of the manuscript.

Validity of the findings

My greatest concern regarding the entire paper is in the statistical methods. I consider the failure to adjust for multiple comparisons inappropriate and misleading. Specifically, under “shear force data analysis,” it is stated that a t-test between means was done (presumably only for those where a p<0.05 was found for the group, although this should be stated). Using a t-test for multiple comparisons greatly increases the likelihood of a type I error. As an example, the p-value for dressing percentage is 0.05. It is difficult to explain why the treatments would notably impact dressing percentage, and this p-value is “marginal.” It would be expected in such cases that post-hoc pair-wise comparisons would not find a significant difference between most (or possibly all) pairs. That isn’t the case when an unadjusted t-test is used, resulting in (likely capricious) “significant differences” between the various groups. It is unclear, but possible that an even more notable statistical error was made regarding table 7. If the overall p-value is 0.08, then no post-hoc between treatment comparisons should be done. Post-hoc pairwise comparisons should only be done when initial analysis finds a statistically significant difference between treatments, and under that circumstance, pairwise comparisons must account for the increased possibility of an error due to multiple comparisons.

It is unclear what the definition of “outlier” is, and thus exclusion of individual animals from various analyses seems arbitrary. This should be clarified, and ensure that consistent standards are applied in all cases. Particularly given the complexity of pharmacokinetic data, it is difficult for a reader to determine whether outliers truly represent an anomaly they are unlikely to deal with, or whether the outliers occurred due to unknown variability with the delivery method. If it is the latter, then the practitioner should know that darting can produce such variability, and that values obtained in practice will share this variability.

Reporting of results in the table also gives me concern. I am not familiar with previous studies examining ceftiofur metabolism, and thus I am uncertain of number of subjects and range of variability commonly seen. However, given the extreme variability and the relatively small sample size in the present study, I question the appropriateness of presenting mean values for Cmax, and comparison of those values to ones reported previously. There is a similar concern would exist for AUMC parameters (where the coefficient of variation is >50%). Indeed, a median value is reported for Tmax, but no explanation is provided as to justification/rationale for reporting median here but not the other parameters. While I concede that inclusion of individual animal data helps the reader to interpret some of this, I believe it would likely be worthwhile to include median values. Discussion should also be offered as to the appropriateness of relying upon mean values as opposed to medians (particularly vis a vis the expected variation, given results from other studies).

Additional comments

I am in favor of acceptance of the manuscript, if authors address concerns noted above. Of particular concern are issues related to statistical methods, and reporting of results.

---

## Round 0.2 · Major Revisions

Thank you for addressing the reviewers and my comments. Experts have reviewed your revised manuscript and while several important comments have been addressed, I invite you to respond to their updated review.

Reviewer 1 ·

Basic reporting

The manuscript is considerably improved from the first version; however, there are still 5 outstanding items to address )Items 1, 2, 3, 5, and 6), as well as one new comment (Item 4). These are described here and/or in the following sections as appropriate, with the Original Reviewer Comment, the Author Reply, and then New Comments from the Reviewer.


1. Original Reviewer Comment - Since the study design does not allow for direct comparison between conventional and dart delivery injection methods, the conclusions should avoid comparative terms such as “altered”, or should state that meat tenderness/pharmacokinetic data may be altered when compared with existing data from previous studies.

Author Reply - This is a fair point, so we’ve revised the conclusion.

New Comments from the Reviewer - The abstract should contain a similar edit, as it is unchanged from the original version. In the reviewer’s opinion, a more appropriate conclusion would be to simply state that this study describes the pharmacokinetic properties of four drugs administered by a pressure-adjustable pneumatic gas-powered dart gun, thus avoiding comparative statements entirely.

2. Original Reviewer Comment - The format for reporting p values should be consistent throughout
o Italicized vs normal text
o Spaces between the p, the sign, and the number vs no spaces
o Same number of significant digits throughout

Author Reply - Changes made as suggested.

New Comments from the Reviewers - Some changes have been made but there are still some inconsistencies. P-values are sometimes given as an exact number (p=0.003, line 271) and sometimes given as a “less than” or “greater than” (p>0.05, line 268). Either format is appropriate, but it should be consistent. It is generally accepted to report exact p-values for all results except those below 0.001, in which case p < 0.001 is appropriate.
Sometimes the p-value is capitalized (line 258) and sometimes it is lowercase (line 268). This should be consistent. Sometimes the symbol p is directly adjacent to the greater-than sign (line 268) and sometimes there is a space between them (line 274). The format should be consistent.

3. Original Reviewer Comment - Would it be possible to combine tables 6 and 7, maybe even table 5?

Author Reply - I combined the tables and hopefully these look better.

New Comments from the Reviewers - Most of the data from Tables 6 and 7 have been added to Table 5 – it looks much better. Table 7 is now redundant and should be removed from the manuscript. Table 6 has some data on position of the bottom-round steaks that can also be moved to Table 5, eliminating the need for table 6 as well.

The column names “Control” and “Darted” in Table 5 appear to be associated with wrong W-B SF values below them (i.e., they have been switched during reformat of the tables).

The explanation of the positions of the bottom-round steaks on the darted and non-darted side in the methods section has improved greatly from the original version, as has the position of the top loin steaks from the anterior edge (presumably referred to as positions 1, 2, 3, and 4 in Table 5). It would add clarity to eth manuscript to include reference to positions 1, 2, 3, and 4 in the materials and methods, as well as footnotes to Table 5 to clearly define/describe the positions of both the bottom-round and top loin steaks, including that “Dart Side” actually refers to the presumed dart entry site on the dart side.

4. New Comment form the Reviewer - Some drugs/equipment sources are not cited completely (the company name is given, but not the city and state where the company is located).

Experimental design

5. Original Reviewer Comment - The methods are consistent with ethical standards in bovine veterinary research; however, the illegal use of CCFA should be more clearly addressed as to why it was included in the study, reiterate the fact it’s illegal, describe whether these animals entered the food supply, etc.

Author Reply - Language about legality has been added to the methods and discussion sections.

New Comments from the Reviewers - The revised manuscript adequately emphasizes that CCFA use is illegal when administered by an unapproved route in the United States, but we still have the following concerns:

- The authors have stated that the animals given CCFA did not enter the food supply in their rebuttal letter, but we were unable to find any point in the revised manuscript where this was stated. This should be clearly stated in the Abstract, Materials and Methods, Discussion, and Conclusions.
- It is still unclear to the reader why CCFA was included in the study. Anecdotal reports that CCFA is sometimes delivered by dart gun illegally in the US are, in our opinion, not good reasons to study the pharmacokinetics and effects on meat quality of the drug when administered by this route. Data obtained from this study on CCFA cannot inform decision-making by producers or veterinarians in the US because the use of the drug by that method would be illegal anyway. The authors should discuss potential applications of their research into this route of administration for CCFA.
- It is important to state precisely what is prohibited by the regulatory restrictions on the use of CCFA. The statement on lines 107-108 that “cephalosporins may not be given in an extra-label manner in the US” is not quite correct. According to the article cited, the FDA order allows for the extra-label use of cephalosporins to treat or control a disease condition that is not present on the label, for example. The relevant prohibition is specific to the use of cephalosporins by an unapproved dose level, frequency, duration, or route. The statements on illegal use of cephalosporins in the manuscript should be revised to make sure they are consistent with this reference.
- The cited article on line 108 should be cited as (Federal Register, 2012) or (US Food and Drug Administration, 2012). Citing it as Anonymous makes the source appear less authoritative.

6. Original Reviewer Comment - The description of the blocking of animals and assignment to treatment groups is well done, however it is unclear to the reader how the final numbers in the pharmacokinetic study were reached. It is evident that more animals were randomly assigned to treatment groups than were analysed - forty steers were randomly assigned, yet only 5-6 per drug treatment, for a total of 17, had blood samples measured. It is unclear how this subset of animals was selected for analysis; were they randomized? Additionally, 3 animals were injected with ceftiofur at a later time, but the reason is not explained in the manuscript. More details should be provided in the methods and results section outlining the criteria used to select animals for analysis, and details should be provided about the origin of the additional 3 animals in the ceftiofur group, as well as the rationale for adding them to the study.

Author Reply - Methods section has been revised to clarify.

New Comments from the Reviewers - The authors should clearly explain the reasons for the apparent removal of data points between revisions. The first draft reads as though 10 animals were blocked and randomly assigned to the CCFA group at the beginning of the study. Five of these animals were included in the pharmacokinetic analysis, and three additional animals were administered ceftiofur at a later time (lines 119-120 of original version) and were also included in the pharmacokinetic analysis. A total of 13 animals appear to have been injected with CCFA in the original version of the manuscript, but only 10 are discussed in the revised version. The apparent exclusion of animals in this section between revisions requires explanation. If additional animals were not added to the study at a later time and the original manuscript was in error, the reason for the error should be clearly explained in the rebuttal letter. If additional animals were added to the study, then the details of the origin of these animals and the rationale for including them in the study should be explained in the revised manuscript.

Validity of the findings

Please see above comments in sections 1 and 2

Additional comments

Please see above comments in sections 1 and 2

·

Basic reporting

Minor recommendations for changes- see below.

Experimental design

No comment.

Validity of the findings

Line 392: It bears acknowledgement that pharmacokinetics, withdrawal times, and clinical efficacy may differ for all three products when delivered via dart as compared to conventional delivery (as you’ve done). Nonetheless, I find it notable that tildipirosin had much less extreme variability in several of the pharmacokinetic parameters, compared to that seen with CCFA and tulathromycin. This is most clear for Cmax and Tmax. I suggest you consider addressing this in the discussion or conclusions. If I had to make a decision on drug selection based upon this study, I would choose tildipirosin. Is that decision defensible?

Additional comments

Line 94: Provide manufacturer info. for Zuprevo
Line 99: How was the determination of need for 104 psi made? Is that based upon dart manufacturers’ recommendations, as determined by volume? Or from a previous report? Or based upon trial and error process?
Line 116: standardize phrasing and punctuation for the groups (there’s a dash after “ceftiofur, but not after tulathromycin).
Lines 254-255: “…dressing percentage and L* values were significant sources of variation within treatment.” This would generally mean that, when inserted as a covariable, dressing percentage and L* values altered the impact of treatment on a third variable. From examining table 4, I do not believe that is what is meant by this sentence. My interpretation of table 4 is that “…dressing percentage and L* values were significantly different between treatment groups.” Or “…dressing percentage and L* values differed significantly across treatment groups.” Or “….significant differences in dressing percentage and L* values were found among treatments.”
Line 256: I’m surprised that an ANOVA value for the full group comparison can be exactly equal to the value found after post-hoc correction for multiple comparisons. It is also surprising that the p-value is exactly the same for two post-hoc comparisons (SALN vs. CCFA and SALN vs. TILD, particularly considering the mean dressing % varied by 0.5% between CCCFA and TILD. Please confirm these values prior to publication.
Line 296: Eliminate double period punctuation.
Line 392: It bears acknowledgement that pharmacokinetics, withdrawal times, and clinical efficacy may differ for all three products when delivered via dart as compared to conventional delivery (as you’ve done). Nonetheless, I find it notable that tildipirosin had much less extreme variability in several of the pharmacokinetic parameters, compared to that seen with CCFA and tulathromycin. This is most clear for Cmax and Tmax. I suggest you consider addressing this in the discussion or conclusions. If I had to make a decision on drug selection based upon this study, I would choose tildipirosin. Is that decision defensible?

---

## Round 0.3 · Minor Revisions

Thank you for addressing the reviewer comments and edits, there seems to be few more edits before the manuscript can be accepted. Please respond to the reviewer comments and I look forward to seeing your revised manuscript.
Best wishes,
Sharif

Reviewer 1 ·

Basic reporting

No further comments (please see "general comments" below).

Experimental design

No further comments.

Validity of the findings

No further comments.

Additional comments

Thank you very much for your changes and explanations in the letter. The manuscript is markedly improved from its earlier versions and is suitable for publication with some recommendations, outlined below. We appreciate the improvements to the methods section and clarification of the experimental numbers. These are much clearer in the manuscript’s current version.

We understand the authors’ concern about excessive repetition of the point about the illegality of extralabel use of CCFA, however it is important to include a short statement in the abstract, since some readers will only read this section. A sentence in the methods section of the abstract, perhaps on line 32 (tracked changes document), that states: “Due to the prohibition of extralabel injection routes for CCFA in the United States, the animals injected with this drug via dart gun did not enter the food supply.”, or a similar sentence to that effect, should be used.

We also understand the rationale for using P>0.05 in some cases rather than the exact P-values. For consistency, they should be written as P>0.050 (with the same number of significant figures as the exact P-values reported elsewhere in the text).

Some brief copy-editing items are as follows. The line citations refer to the tracked changes document:
o Line 222 “top round”: We think this should read “bottom round”.
o 9 animals were injected with ceftiofur according to figure 1, but only 8 were injected in this group in the text and in table 1.
o 6.6mg/kg dose of ceftiofur stated in figure 1/table1, but only 6mg/kg was stated in the text.
o Line 39 on table 5 should read “text” not “test”.
o Table 6 and 7 are still cited in the discussion section after their removal.
o The P-value on line 279 is differently formatted than the rest of the P-values in the manuscript.
o The citation of Anonymous (the first reference) has changed in the reference section as per our recommendation but the in-text citations throughout the manuscript have not changed.

Thank you very much for your attention to these edits.

·

Basic reporting

Line 76: “…important contributor to the consumer….” seems like an odd (and inaccurate) phrasing. Suggest: “…. Important contributor to consumer satisfaction…” or “important characteristic for the consumer….”
Line 117: “….were administered ceftiofur…” Specify the formulation was CCFA. Also specify route- was this administration per label, or administered in the round (to better compare to that administered via dart)?
Line 271: Move “(P=0.73)” from after “affected” to between “…loin” and “affected….”
Line 307: Should this be “similar” rather than “equivalent?”
Line 379: “…the ones that did…” could be modified to state “….the ones that penetrated the skin….”
Line 275 states “all steaks (from darted and non-darted sides) from cattle treated with ceftiofur and saline were more tender than from cattle treated with tulathromycin or tildipirosin.” Could this not possibly indicate a difference in the cattle themselves more so than being attributable to treatment (considering it was also seen in the non-darted sides)? This possibility isn’t mentioned (although admittedly, the discussion states that the difference may not be clinically significant).
Table 1: Tmax mean has a value of 8*, with the asterisk indicating median rather than mean. It would seem more appropriate to me to not include any value there (simply an asterisk saying “median is preferred” or similar) rather than putting an inaccurate value with an asterisk that it isn’t what it states it is.
Table 1: The legend/key refers to t1/2, but the table has t½λz. These should be the same.
Table 5: I would suggest changing layout for the p-values to be positioned elsewhere, but will defer to the editor.

Experimental design

The number of cattle enrolled in the ceftiofur treatment is inconsistent. The abstract states 8 (line 30); materials and methods states 5 (line 115) immediately followed by “Three additional steers….” (so total of 8? Although it isn’t clear that these three were handled the same way). Figure 1 legend states “Mean serum concentrations of ceftiofur…. In 9 animals” and table 1 shows results from 8 individuals.

Validity of the findings

No comment

---

## Round 0.4 · accepted · Accept

Thank you for addressing the reviewers' comments, I found your manuscript acceptable for publications. Congratulations and best wishes in your research.

Sharif